# Deep-Eutectic-Solvent-Assisted Synthesis of a Z-Scheme BiVO_4_/BiOCl/S,N-GQDS Heterojunction with Enhanced Photocatalytic Degradation Activity under Visible-Light Irradiation

**DOI:** 10.3390/mi13101604

**Published:** 2022-09-27

**Authors:** Hengxin Ren, Kuilin Lv, Wenbin Liu, Pengfei Li, Yu Zhang, Yuguang Lv

**Affiliations:** 1College of Pharmacy, Jiamusi University, Jiamusi 154007, China; 2College of Materials Science and Engineering, Jiamusi University, Jiamusi 154007, China; 3China Building Materials Test and Certification Group Co., Ltd., Chaoyang District, Beijing100024, China

**Keywords:** BiVO_4_, BiOCl, S, N-GQDS, heterojunction, photocatalysts

## Abstract

Z-scheme heterojunction photocatalytic nanomaterial designs have attracted attention due to their high catalytic performance. Deep eutectic solvents (DESs) have been used as green, sustainable media, acting as solvents and structure inducers in the synthesis of nanomaterials. In this work, a novel visible-light-absorption-enhanced bismuth vanadate/bismuth oxychloride/sulfur, nitrogen co-doped graphene quantum dot (BiVO_4_/BiOCl/S,N-GQDS) heterojunction photocatalyst was prepared in a deep eutectic solvent. The photosynthetic activity of the BiVO_4_/BiOCl/S,N-GQDS composite was determined by the photocatalytic degradation of rhodamine B (RhB) under visible-light irradiation. The results showed that the highest photocatalytic activity of BiVO_4_/BiOCl/S,N-GQDS was achieved when the doping amount of S,N-GQDS was 3%, and the degradation rate of RhB reached 70% within 5 h. The kinetic and photocatalytic cycles showed that the degradation of Rhb was in accordance with the quasi-primary degradation kinetic model, and the photocatalytic performance remained stable after four photocatalytic cycles. Ultraviolet–visible diffuse reflectance (UV-DRS) and photoluminescence (PL) experiments confirmed that BiVO_4_/BiOCl/S,N-GQDS ternary heterojunctions have a narrow band gap energy (2.35 eV), which can effectively improve the separation efficiency of the photogenerated electron–hole pairs and suppress their complexation. This is due to the construction of a Z-scheme charge process between the BiVO_4_/BiOCl binary heterojunction and S,N-GQDS, which achieves effective carrier separation and thus a strong photocatalytic capability. This work not only provides new insights into the design of catalysts using a green solvent approach but also provides a reference for the study of heterojunction photocatalytic materials based on bismuth vanadate, as well as new ideas for other photocatalytic materials.

## 1. Introduction

In recent years, with population growth, sustained industrial development and economic globalization, human consumption of fossil energy has accelerated and released large quantities of organic and toxic pollutants into the natural environment, causing serious ecological damage. Traditional water control technology has the problems of cumbersome treatment methods and high application costs. Therefore, it is necessary to develop a practical, effective and environmentally friendly water treatment technology [1]. 

Since 1978, when J. H. Carey et al. used TiO_2_ to catalyze the oxidative degradation of polychlorobiphenyls, photocatalysis has attracted a great deal of interest from researchers as an advanced technology for the oxidative degradation of pollutants [2]. Photocatalytic degradation based on semiconductor photocatalysts is deemed to be a very promising technology for environmental remediation due to its energy savings, environmental friendliness and high degradation efficiency. In recent years, researchers have conducted many explorations to study the synthesis, application and catalytic mechanisms of semiconductor photocatalysts [3]. Nanosemiconductors, including BiVO_4_, ZnS, CdS, Fe_2_O_3_, ZnO and Ag_2_WO_4_, have been used for a variety of environmental purifications, such as organic degradation, microbial inactivation and water splitting [4]. Despite their effectiveness, they suffer from e-/h+ recombination, wide-band-gap energy, and lower stability, all of which can significantly reduce their efficiency [5]. Therefore, some important techniques and modifications, such as the preparation of nanoparticles, doping and the construction of heterojunctions, have been applied to address these drawbacks [6].

N-type monoclinic crystalline BiVO_4_ has attracted broad attention because of its narrow band gap (about 2.40 eV), so it has the advantages of visible-light-driven activity, a wide light absorption range and good photocatalytic performance. It is considered to be the most promising of the semiconductor photocatalytic reaction materials. It has been shown that BiVO_4_ can better catalyze the degradation of organic pollutants under visible-light irradiation, but its photocatalytic degradation performance is limited due to the photogenerated carrier’s low migration efficiency and fast recombination [7]. BiOCl is a p-type semiconductor material in which the chlorine atoms between the [Bi_2_O_2_] layers are staggered along the c-axis to form a two-dimensional nano-layered structure, which facilitates the separation and transfer of the photogenerated carrier. However, due to the large band gap (about 3.5 eV), BiOCl absorbs less than 5 % of the UV light in sunlight and thus has poor photocatalytic performance under visible light [8]. It is worth noting that when a p-type semiconductor and an n-type semiconductor form a p-n heterojunction, an internal electric field will be formed between them, which accelerates the transfer of the photogenerated carrier and improves the separation efficiency of electron–hole pairs. This serves as a strategy to improve the activity of the BiVO_4_ photocatalyst [9]. At present, BiOCl/BiVO^4^ p-n heterojunction photocatalysts are synthesized by various methods, and their band gaps have been adjusted to increase their photoabsorption range and enhance their photocatalytic degradation capability for organic pollutants [10].

Sulfur–nitrogen-co-doped graphene (S,N-GQDS) has attracted much attention for its good biocompatibility, conductivity and photochemical properties [11]. These unique properties can be utilized to construct ternary heterojunctions with BiOCl and BiVO_4_. When S,N-GQDS constructs a heterojunction with BiOCl and BiVO_4_, it will be in close contact with BiOCl and BiVO_4_. This will accelerate the interfacial charge transfer rate of the BiVO_4_/BiOCl/S,N-GQDS heterojunction, reduce the impedance effect and enhance its photocatalytic properties. Fortunately, the structure of the ternary BiVO_4_/BiOCl/S,N-GQDS heterojunction photocatalyst has not been reported [12]. 

The synthesis of a ternary heterojunction is usually difficult and requires expensive and even poisonous templates or surfactants, as well as strict synthetic conditions. A deep eutectic solvent (DES) is a kind of non-toxic, cheap and completely biodegradable organic solvent. By changing the charge neutralization, reduction potential or chemical activity, and passivation in a given crystal plane, the composition of the DES can modify the nucleation and growth mechanism of the crystal and thus determine its growth direction. Therefore, it is particularly suitable for the large-scale synthesis of novel functional materials, and as a pure green solvent, it is receiving more and more attention [13].

In summary, a novel BiVO_4_/BiOCl/S,N-GQDS ternary heterostructure was prepared by the hydrothermal method with a deep eutectic solution as the solvent system. The photocatalytic degradation mechanism of the BiVO_4_/BiOCl/S,N-GQDS heterojunction was evaluated using rhodamine B (RhB) as the model pollutant. This research is of great significance for solving the problem of water pollution.

## 2. Experimental Section 

### 2.1. Photocatalyst Preparation 

Preparation of bismuth vanadate: A mixture of choline chloride and urea (1:2) was heated at 70 °C for 30 min to produce 2 g of DES. One gram of distilled water was added to the solution to adjust its viscosity. Then, 0.01 mol Bi(NO_3_)_3_·5H_2_O and 0.03 mol NH_4_VO_3_ were dissolved in 30 ml of DES and stirred by magnetic force for 1 h. The suspension was transferred to a 100 mL Teflon-lined stainless autoclave and heated at 180 °C for 5, 10 or 15 h. The product was cleaned repeatedly with deionized water and anhydrous ethanol, dried in a vacuum-drying box at 60 °C for about 12 h and kept in a Muffle Furnace at 500 °C for 2 h. The yellow BiVO_4_ sample was finally obtained. In addition, 0.01mol Bi (NO_3_)_3_·5H_2_O was added to a 100mL HCl solution (0.5 mol·L^−1^) and ultrasonicated for 0.5 h. The reactants were placed in an autoclave and reacted at 180 °C for 15 h. BiOCl samples were obtained by the same cleaning and drying methods.

Preparation of S,N-GQD_S_: First, 1.26 g of citric acid and 1.38 g of thiourea were dissolved in 30 mL of deionized water, stirred for 1 h to obtain a clear solution and then transferred to a 100 mL Teflon-lined stainless autoclave and heated at 180 °C for 8 h. The solution was cooled to room temperature and centrifuged at a speed of 10,000 rpm for 15 min. The precipitates were washed with deionized water and anhydrous ethanol (95%) several times and freeze-dried to obtain reddish-brown S,N-GQD_S_ solids. 

The preparation of BiVO_4_/BiOCl/S,N-GQD_S_:1mmol BiVO_4_ was dispersed in 10 mL of a HCl (0.5 mol L^−1^) solution and stirred by magnetic force for 1 h. Then, precursors with different qualities were added and stirred for 0.5 h. The solution was transferred to a 25 mL Teflon-lined stainless-steel autoclave and heated at 180 °C for 10 h. The reaction products were washed with deionized water and ethanol many times and dried in a vacuum-drying box at 60 °C for 12 h. The samples were obtained and recorded as BiVO_4_/BiOCl/S,N-GQD_S_ (1%), BiVO_4_/BiOCl/S,N-GQD_S_ (3%) and BiVO_4_/BiOCl/S,N-GQD_S_ (5%) according to the different mass fractions of S,N-GQD_S_ in the samples. In addition, BiVO_4_/BiOCl samples were obtained by using the same treatment method mentioned above without doping the precursors of S,N-GQD_S_.

### 2.2. Photocatalytic Characterization

X-ray diffraction measurements were made by using a Bruker D8 Advance X-ray diffractometer (Bruker Aus, Germany) with Cu Kα radiation in the range 10° ≤ 2θ ≤ 70°. Fourier transform infrared (FT-IR) spectra were recorded on an IR Prestige-21 spectrometer (Shimadzu, Kyoto, Japan) using the KBr pellet technique at room temperature. An S-4700 Field Emission Scanning Electron Microscope (SEM; Hitachi Co., Japan) and a JEM-2010 transmission electron microscope (TEM; JOEL Ltd, Tokyo, Japan) were used to characterize the structure and morphology of the products. The light response ability of the sample was measured over a wavelength range of 800–200 nm on a UV-4100 UV-vis spectrophotometer (Shimadzu, Japan). The element composition and status of the sample were measured using X-ray photoelectron spectroscopy (Thermo ESCALAB 250XI, Thermo Fisher, Waltham, MA, USA). Photoluminescence (PL) spectra were measured using a Hitachi F-4600 fluorescence spectrophotometer (Hitachi, Tokyo, Japan) equipped with a 450 W Xenon lamp as the excitation source.

### 2.3. Evaluation of Photocatalytic Performance

The photocatalytic degradation of photocatalysts was evaluated by using rhodamine B as an organic dye model. First, 0.05 g of the photocatalyst sample was added to a 50 mL 10 mg·L^−1^ rhodamine B solution and stirred by magnetic force in the dark for 30 min. After the adsorption was balanced, a xenon lamp of 350 watts was irradiated as the visible light source. The absorption of a 5 mL solution was measured at intervals of 30 min. After centrifugation at high speed, the solution was purified, and the maximum absorption wavelength of 554 nm was determined by a UV-vis spectrophotometer. Finally, the solution was poured back into the double beaker, and the reaction continued. The calculation method is given in Formula (1).
(1)$$\mathrm{Degradation}\;\mathrm{rate}\;\mathrm{of}\;\mathrm{RhB}=\frac{C_{t}}{C_{0}}\times 100\%$$
where *C*_0_ indicates the initial mass concentration of RhB (mg·L^−1^), and *C_t_* represents the concentration of RhB at different times after photocatalytic degradation. After photocatalytic degradation, the photocatalyst solids were recovered. Subsequently, the photocatalyst was repeatedly cleaned with deionized water and anhydrous ethanol solution and vacuum-dried at 70 °C. Using the same photocatalytic degradation process, four experiments of photocatalytic degradation were carried out, and the X-ray diffraction patterns before and after photocatalytic degradation were compared to investigate the stability of the photocatalyst samples during photocatalytic degradation. 

### 2.4. Evaluation of Photocatalytic Mechanism

The light absorption characteristics of the samples were studied by solid UV-Vis DRS. This method is used to analyze the displacement of the side band and the change in the forbidden band gap between them. The band gap of the sample can be calculated by the formula below.
(2)$$\alpha h\upsilon =A(h\upsilon -E_{g}{)}^{n/2}$$
where *α* is the absorption coefficient, *hυ* is the photon energy, A is a constant equal to 1, and *E_g_* is the band-gap energy. The values of n are 4 for indirect transition and 1 for direct transition [14]. 

The valence- and conduction-band potentials of BiOCl and BiVO_4_ can be calculated using their electronegativity with the following empirical equations.
(3)$$E_{VB}=X+0.5E_{g}-E_{e}$$
(4)$$E_{CB}=E_{VB}-E_{g}$$

Here, *E_CB_* and *E_VB_* represent the conduction band edge and valence band edge, and *E_g_* is the band-gap energy. *X* is the absolute electronegativity of the semiconductor, and *E*_e_ is a measurement scale factor of the redox level of the reference electrode relative to the absolute vacuum scale (−4.5 eV) [15].

In addition, the separation efficiency of photoelectrons and holes in visible-light samples was tested by photoluminescence (PL) spectroscopy. At the same time, iso-Propyl alcohol (IPA), p-Benzoquinone (BQ) and Ethylene Diamine Tetraacetic Acid (EDTA) were used as entrapment agents to detect the main reaction groups in the photocatalytic process. 

## 3. Results and Discussion

### 3.1. Material Characterizations

As shown in Figure 1, all XRD characteristic diffraction peaks belong to BiVO_4_ (JCPDS No. 83-1699) and BiOCl (JCPDS No. 73-2060), indicating the presence of BiOCl and BiVO_4_ in the samples [16]. The diffraction peaks of S,N-GQDs are not obvious, mainly because of their instability in the HCl solution, resulting in a doping ratio lower than the theoretical value. In addition, there were no diffraction peaks of other substances in XRD mode, indicating the high purity of the prepared samples. The diffraction peaks at 2θ = 24.2° and 25.8° of the monoclinic BiVO_4_ samples can be clearly observed when the processing times are 5 h and 10 h. These belong to BiOCl. Meanwhile, the diffraction peaks of the sample at 2θ = 18.6°, 18.9°, 28.5°, 28.8° and 30.5° can be observed. These peaks then belong to the monoclinic crystal system BiVO_4_. This indicates that the sample is a mixture of BiOCl and monoclinic BiVO_4_. After 15 h of treatment, the sample was in the monoclinic pyroxene BiVO_4_ pure phase, crystallization was good, and the corresponding BiOCl diffraction peaks disappeared. This is due to hydrogen bonding between the hydroxyl group of choline chloride and the polar group of urea, as well as unsaturated bonds and hydroxyl groups in the BiVO_4_ crystal nucleus, which cause some solvent molecules to adsorb on the surface of BiVO_4_, thus inducing the formation of BiOCl [17]. The characteristic peaks of BiVO_4_ at 2θ = 18.98° and 30.52° correspond to the {011} and {112} crystal planes, respectively. When BiVO_4_ is recombined with BiOCl or S,N-GQD_S_, their intensity decreases noticeably, revealing that BiOCl and S,N-GQD_S_ are favored to recombine with BiVO_4_ on these two crystal planes. The crystallinity of the BiVO_4_ sample decreased, and the half-peak width increased after compounding, which indicates that BiOCl and S,N-GQD_S_ inhibit the growth of the monoclinic BiVO_4_ crystal.

As shown in the TEM image of the BiVO_4_ sample in Figure 2A, the BiVO_4_ sample is a block crystal. In Figure 2B, crystals in the BiVO_4_/BiOCl composite sample can be found stacked and nested together. Figure 2C demonstrates that the distribution of S,N-GQD_S_ is scattered without obvious aggregation. In Figure 2D, BiVO_4_, BiOCl and S,N-GQD_S_ are tightly bound together, from which different lattice stripes can be observed. The part with the lattice spacing of 0.374 nm is consistent with the {110} crystal plane and the lattice of BiOCl [18]. The part with the lattice spacing of 0.290 nm is attributed to the {011} crystal plane of BiVO_4_, which is consistent with the XRD analysis [19]. In addition, the lattice spacing of S,N-GQD_S_ is 0.241 nm, similar to the striations of the {1120} crystal plane of graphene [20]. This indicates that S,N-GQD_S_ has the properties of graphene, which forms a ternary heterostructure together with the BiVO_4_/BiOCl p-n heterojunction.

It can be clearly observed from the SEM images that a large number of BiOCl nanosheets were attached to the surface of bulk BiVO_4_ when the treatment time was 5 h (Figure 3A), while the BiOCl nanosheets gradually decreased when the treatment time was extended to 10 h (Figure 3B). Finally, when the treatment time reached 15 h, the BiOCl nanosheets disappeared, and only the pure-phase bulk BiVO_4_ was retained (Figure 3C).

This is mainly due to the induction of the deep eutectic solution during the synthesis of BiVO_4_, which can lead to the morphological evolution of BiVO_4_ samples with the increase in retention time. The XRD characteristic analysis results confirm this phenomenon [21]. As seen in the SEM images (Figure 3D) of BiVO_4_/BiOCl samples with a high density of lamellar BiOCl and bulk BiVO_4_ materials superimposed on each other, the formation of the BiVO_4_/BiOCl p-n heterojunction interface facilitates the transfer of photogenerated charges [22]. As shown in Figure 3E, in the BiVO_4_/S,N-GQD_S_ sample, a large amount of bulk BiVO_4_ is visible, and only a small amount of spherical S,N-GQD_S_ is distributed in it, which is related to the low doping of S,N-GQD_S_. As shown in Figure 3F, the BiVO_4_/BiOCl/S,N-GQD_S_ sample consists of high-density BiOCl flakes and bulk BiVO_4_ material stacked together, while S,N-GQD_S_ are dispersed in it to assemble heterojunctions.

The EDS energy spectrum is shown in Figure 4, and the BiVO_4_/BiOCl/S,N-GQD_S_ (3%) composite contains Bi, O, V, S, N, C and Cl elements, further confirming the coexistence of BiVO_4_, BiOCl and S,N-GQD_S_. The calculated results show that the V:Cl atom percentage is 1:0.081, which is slightly lower than the theoretical mole ratio of 1:0.1. The Bi:(V+Cl) atomic percentage is 1.08:1, close to the theoretical value of the BiVO_4_/BiOCl complex of 1:1. The weight percentage of BiVO_4_:S,N-GQD_S_ is 1:0.026, which is close to the 3% doping ratio. The above results demonstrated that BiVO_4_/BiOCl/S,N-GQD_S_ (3%) were synthesized successfully.

Figure 5A shows the full spectrum of the X-ray photoelectron spectra of BiVO_4_/BiOCl/S,N-GQD_S_ (3%) photocatalyst samples. It consists of Bi, V, Cl, O, S, N and C elements with atomic mass percentages of 5.15, 3.57, 1.17, 30.01, 0.31, 4.52 and 55.27, respectively. Figure 5A shows the X-ray photoelectron spectroscopy results of samples of the BiVO_4_/BiOCl/S,N-GQD_S_ (3%) photocatalyst. It includes Bi, V, Cl, O, S, N and C elements with atomic mass percentages of 5.15, 3.57, 1.17, 30.01, 0.31, 4.52 and 55.27, respectively. The calculated atomic percentage of bismuth:(V/Cl) is 1.09:1, which is close to the theoretical 1:1 value of the BiVO_4_/BiOCl complex. This is consistent with the results of the EDS analysis. In Figure 5B, the XPS spectrum of C1s shows a dominant peak at a binding energy of 284.95 eV, which is attributed to C-C due to sp2 hybridization. In addition, there are two characteristic peaks of lower intensity. The characteristic peak with a binding energy of 286.65 eV belongs to C-N, C-S and C-O due to sp3 hybridization. The characteristic peak with a binding energy of 288.65 eV belongs to C=O in carbonyl and carboxylates [23]. As shown in Figure 5C, the characteristic peaks of binding energies at 529.5 eV, 531.8 eV and 533.2 eV are high-resolution XPS signals of O 1s, corresponding to Bi-O and absorbed oxygen on the composite surface, respectively [24]. The two strong peaks of binding energies at 164.2 eV and 159.0 eV in Figure 5D are Bi 4f5/2 and Bi 4f7/2, respectively, which are characteristic peaks of Bi^3+^. This indicates that the type of bismuth in the samples of BiVO_4_/BiOCl/S,N-GQD_S_ (3%) is Bi^3+^. The characteristic peaks of V^5+^ at 516.1 and 523.8 eV in Figure 5E are attributed to V 2p3/2 and V 2p1/2, respectively. The characteristic peaks of Cl 2p3/2 and Cl 2p1/2 can be observed at 197.8 eV and 199.4 eV (Figure 5F) [25]. The XPS signals of N1s can be seen at 398.5, 400.8 and 402.2 eV, attributed to imine (=N^−^), amine (-NH-) and protonated imine (=NH^+^), respectively. In addition, the S2p spectrum in Figure 5H clearly shows the characteristic peak of S^2−^ at 224.63 eV. The presence of the characteristic peaks of S2p and N1s confirms that S and N elements were successfully doped into GQD_S_ [26]. The coexistence of BiVO_4_, BiOCl and S,N-GQD_S_ in the composite was further confirmed by XPS, XRD and SEM. 

The Fourier transform infrared spectroscopy (FT-IR) spectrum of the sample is shown in Figure 6. Absorption peaks at 512 cm^−1^ all appear due to the stretching vibration of Bi-O, while the absorption peak at 742 cm^−1^ belongs to the bending vibration of *v*3 (VO_4_). The wavelength near 1622 cm^−1^ belongs to the stretching vibrational band of C-O in the -COOH and -CONH groups of S,N-GQD_S_ [27]. The distinct vibrational peak at 3459 cm^−1^ corresponds to the O-H stretching vibration of the water molecule. Due to the small doping amounts of S and N elements and the strong stretching vibration of VO_4_^3−^ and Bi-O, there are no obvious C-N and C-S absorption peaks [28]. In summary, it can be concluded that BiVO_4_/BiOCl/S,N-GQD_S_ composites contain Bi-O and VO_4_^3−^ groups. 

### 3.2. Photodegradation Properties

Under simulated-natural-light irradiation, rhodamine B dye was used as the target contaminant for degradation. The photocatalytic degradation activities of BiVO_4_ samples prepared with different holding times, namely, BiVO_4_/BiOCl, BiVO_4_/S,N-GQD_S_ complexes and BiVO_4_/BiOCl/S,N-GQD_S_ complex samples with different doping masses of S,N-GQD_S_, were analyzed. In order to accurately determine the concentration of rhodamine B solution, the standard curve of the rhodamine B (RhB) solution was established using different concentrations of rhodamine B standard solutions, and the regression equation and correlation coefficient R^2^ were obtained by linear fitting. As shown in Figure 7A, the equation of the standard curve of the rhodamine B (RhB) solution was: y = 0.2015x − 0.0312, and R^2^ = 0.9991, indicating that the equation was quite linear and could be used for the determination of the rhodamine B solution concentration. The photocatalytic degradation curves of each sample are shown in Figure 7B. The photocatalytic degradation rates in order of strength to weakness within 5 h are: BiVO_4_/BiOCl/S,N-GQD_S_ (3%) (85%) > BiVO_4_/BiOCl/S,N-GQD_S_ (1%) (81%) > BiVO_4_/BiOCl/S,N-GQD_S_ (5%) (78%) > BiVO_4_/BiOCl (62%) > BiVO_4_/S,N-GQD_S_ (59%) > BiVO_4_-10h (55%) > BiVO_4_-15h (46%) > BiVO_4_-5h (40%). To quantitatively describe the rate of the catalytic reaction, the kinetic fitting of the reaction of the material to degrade the target contaminant was performed by fitting the above degradation curve with the Langmuir–Hinshelwood equation. The calculation method is shown in Equation (5).
(5)$$\ln(C_{0}/C_{t})=kt$$
where ln is the natural logarithm, and *K* is the photodegradation reaction constant. A larger value of *K* means better photocatalytic activity of the sample. In addition, *C*_0_ is the initial concentration of the RhB dye, and *C_t_* is the RhB concentration at reaction time t. As shown in Figure 7C, the reaction rate constants for different preservation times (5, 10 and 15 h), BiVO_4_/BiOCl, BiVO_4_/S,N-GQD_S_ complexes and BiVO_4_/BiOCl/S,N-GQD_S_ were 0.1565, 15657, 160.47, 130.475, 240.377, 240.377 and 0.267, respectively. Meanwhile, the correlation coefficients for linear fits, R^2^, were determined to be 0.9582, 0.97061, 0.9756, 0.99341, 0.98718, 0.98754, 0.99045 and 0.98347. The information in Figure 7B,C shows that the degradation rates of BiVO_4_ samples with a holding time of 15 h are lower than those with a holding time of 5 or 10 h. This is due to the BiVO_4_ sample containing a small amount of BiOCl at a 5 h or 10 h holding time, thus forming a P-n heterojunction with BiVO_4_, so the effect of catalytic degradation is greater than that of the pure phase of BiVO_4_. The heterostructure of BiVO_4_ with BiOCl or S,N-GQD_S_ can significantly improve the photocatalytic degradation, among which BiVO_4_/BiOCl/S,N-GQD_S_ (3%) can obtain the best photocatalytic degradation performance. Figure 7D shows the spectra of 3% BiVO_4_/BiOCl/S,N-GQD_S_ samples at different degradation times. The results showed that, with the prolongation of the degradation time, the absorption value of the RhB solution gradually decreased, the maximum absorption spectra were blue-shifted, and a new absorption peak was produced at 515 nm. This is due to the reaction of RhB with the photocatalyst, which removes a portion of the ethyl groups and generates a new product [29]. 

The stability and reproducibility of the experimental results for the photocatalytic degradation of BiVO_4_/BiOCl/SN-GQDs (3%) catalyst samples are shown in Figure 8. After four photocatalytic cycles, the photocatalytic activity of the BiVO_4_/BiOCl/S,N-GQDs (3%) photocatalyst sample was still high, and the crystal structure was still monoclinic scheelite without any change. This indicates that the BiVO_4_/BiOCl/S,N-GQDS (3%) photocatalyst samples do not undergo photocorrosion during the photocatalytic degradation of RhB and have good stability and reproducibility. The comparative analysis of the performance of the prepared BiVO4/BiOCl/S,N-GQDS samples with other bivo4-based heterojunctions for the degradation of RhB shows that the degradation performance of the prepared BiVO4/BiOCl/S,N-GQDS samples is superior, as shown in Table 1.

### 3.3. Photocatalytic Mechanism

Figure 9A shows the UV–visible diffuse reflectance spectra of the prepared photocatalyst. BiOCl has a light absorption threshold of 310 nm in the UV region. The light absorption threshold of BiVO_4_ and its composite sample was shifted to 500 nm in the visible-light region, indicating the redshift of the sample’s absorption edge after heterojunction formation. As shown in Figure 9B, the forbidden band gap of the BiOCl, BiVO_4_, S,N-GQD_S_, BiVO_4_/BiOCl, BiVO_4_/S,N-GQD_S_, BiVO_4_/BiOCl/S,N-GQD_S_ (1%), BiVO_4_/BiOCl/S,N-GQD_S_ (3%) and BiVO_4_/BiOCl/S,N-GQD_S_ (5%) samples were 3.36, 2.41, 2.1, 2.42, 2.36, 2.38, 2.35 and 2.37 eV, respectively. According to the energy-band theory, the forbidden bandwidth determines the photoresponse range of semiconductor photocatalysts. Obviously, the BiVO_4_/BiOCl/S,N-GQD_S_ (3%) sample has the narrowest forbidden band gap (2.35 eV), so it can utilize more sunlight and enable more photogenerated carriers to separate and transfer, thus improving the photocatalytic performance. The *X* values for BiOCl and BiVO_4_ are about 6.35 eV and 6.03 eV, respectively. Using Equations (3) and (4), the values of *E_VB_* and *E_CB_* of BiOCl were calculated to be 3.53eV and 0.17 eV, while the *E_VB_* and *E_CB_* values of BiVO_4_ were 2.74 eV and 0.33 eV, respectively [34]. The *E_CB_* of S,N-GQD_S_ was determined by XPS to be -0.8 eV, and the *E_VB_* of S,N-GQDS was calculated to be 1.3 eV using Equation (4).

The photoluminescence spectra of BiVO_4_ and its heterojunction photocatalysts are shown in Figure 10, which shows that the photoluminescence spectra of all samples have similar shapes, with emission peaks near 495 and 530 nanometers, respectively. In addition, the decreasing order of the PL peak intensities of these samples is as follows: BiVO_4_-10h > BiVO_4_/BiOCl > BiVO_4_-5h > BiVO_4_/BiOCl/S,N-GQD_S_ (5%) > BiVO_4_-15h > BiVO_4_/BiOCl/S,N-GQD_S_ (1%) > BiVO_4_/S,N-GQD_S_ (3%) > BiVO_4_/BiOCl/S,N-GQD_S_ (3%). Obviously, the PL emission peak intensity of the BiVO_4_/BiOCl/S,N-GQD_S_ (3%) sample is the lowest, which suggests that the construction of the BiVO_4_/BiOCl/S,N-GQD_S_ (3%) ternary heterojunction can effectively decrease the chance of electron and hole complexation through photogeneration.

The results of active-group-capture experiments for the photocatalytic degradation of RhB by BiVO_4_/BiOCl/S,N-GQD_S_ (3%) samples are shown in Figure 11, and it can be found that the addition of EDTA has almost no effect on the performance of the photocatalytic degradation of RhB, but the performance of the photocatalytic degradation of RhB after adding p-benzoquinone and IPA under the same conditions decreased from 85% to 40% and 44% within 300 min, respectively. This indicates that the photocatalytic degradation of RhB in the prepared photocatalytic materials is not performed by the h^+^ reaction group but by the •OH and •O^2−^ reaction groups. 

Based on the above experimental results, Figure 12 introduces in detail the separation and transfer process of the BiVO_4_/BiOCl/S,N-GQD_S_ ternary-heterojunction-photogenerated carrier and its possible photocatalytic degradation mechanism. Since the band gaps of BiVO_4_ and S,N-GQD_S_ are narrow, they can be excited simultaneously under visible-light irradiation. In contrast, BiOCl cannot be excited to produce photogenerated carriers because of its wide band gap. However, this situation changes after the construction of the BiOCl/BiVO_4_/N-GQDs ternary heterojunction. When BiOCl and BiVO_4_ are tightly bonded together, the p-type BiOCl and n-type BiVO_4_ form a BiOCl/BiVO_4_ p-n heterojunction [35]. Meanwhile, an internal electric field is formed at their interface, which can provide a potential driving force to promote electron transfer from the VB of BiOCl to the VB of BiVO_4_, resulting in the generation of holes (h+) in the VB of BiOCl, while no electrons can be generated in the CB of BiOCl. The VB of BiOCl (2.4 eV) is more positive than the redox potential of •OH/OH^−^ (2.38 eV), so h^+^ can oxidize OH^−^ to •OH [36]. Because S,N-GQD_S_ has both p-type and n-type semiconductor characteristics [37], when BiVO_4_ and S,N-GQD_S_ are tightly bound together, the e^−^ in the CB of BiVO_4_ recombines with the h^+^ in the VB of S,N-GQD_S_ through a Z-scheme charge transfer process. This eventually leads to more e^−^ and h^+^ remaining in the CB of S,N-GQD_S_ and in the VB of BiVO_4_, respectively. The CB of S,N-GQD_S_ (−0.80 eV) is more negative than the redox potential of O_2_/•O^2−^ (−0.046 eV), so e- can reduce O_2_ to •O^2^. The oxidative degradation of RhB can be achieved by large amounts of -OH and •O^2−^. On the other hand, RhB consumes more •O^2−^ and •OH during its degradation by photocatalytic oxidation, which induces more electron transfer and generates more e^−^ and h^+^, and so on and so forth until the degradation process is finished.

## 4. Conclusions

A novel visible-light-absorption-enhanced BiVO_4_/BiOCl/S,N-GQD_S_ ternary heterojunction photocatalyst was successfully prepared, and a tentative mechanism to explain the significantly enhanced photoactivity was proposed. The photocatalyst’s catalytic degradation of RhB experimentally proved that the construction of a Z-scheme heterogeneous structure is one of the effective strategies to improve the catalytic activity of the photocatalyst. The highest photocatalytic activity was achieved when the S,N-GQD_S_ doping amount was 3%, and the RhB degradation rate could reach 81%. Moreover, the samples remained stable after four cycles of degradation. The formation of heterojunctions can promote electron transfer and inhibit the complexation of electron–hole pairs, which can significantly improve the photocatalytic activity of pure BiVO_4_. The formation of heterojunctions promotes electron transfer and inhibits the complexation of electron–hole pairs, enhancing the effects of the radicals •O^2−^ and •OH. Meanwhile, the smaller forbidden band gap (2.35 eV) improves the efficiency of visible-light utilization, which can significantly improve the photocatalytic activity of pure BiVO_4_. The economy and environmental friendliness of the solution were improved by the use of deep eutectic solvents. In addition, the issue of how to improve the recovery rate of the material during practical applications while improving the photocatalytic performance of the material is also worthy of our consideration.

## Figures and Tables

**Figure 1 micromachines-13-01604-f001:**
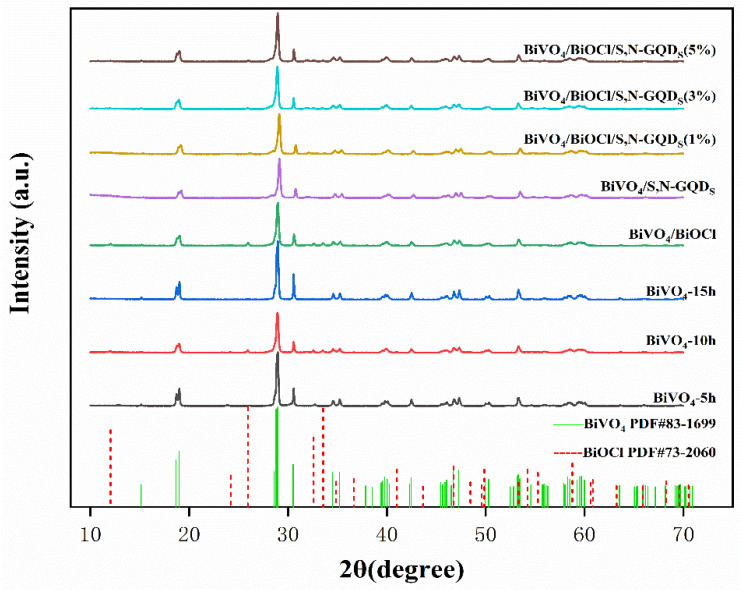
The XRD patterns of BiVO_4_ samples with various treatment times, BiVO_4_/BiOCl and BiVO_4_/S,N-GQD_S_ composites and BiVO_4_/BiOCl/S,N-GQD_S_ complex samples doped with different masses of S,N-GQD_S_.

**Figure 2 micromachines-13-01604-f002:**
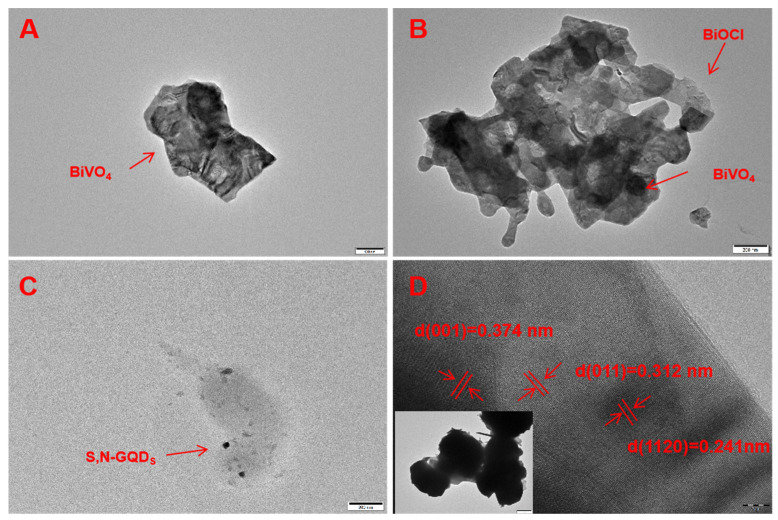
TEM images of BiVO_4_ (**A**), BiVO_4_/BiOCl (**B**) and S, N-GQD_S_ (**C**); TEM and HRTEM of BiVO_4_/BiOCl/S,N-GQD_S_ functional composite nanomaterials (**D**).

**Figure 3 micromachines-13-01604-f003:**
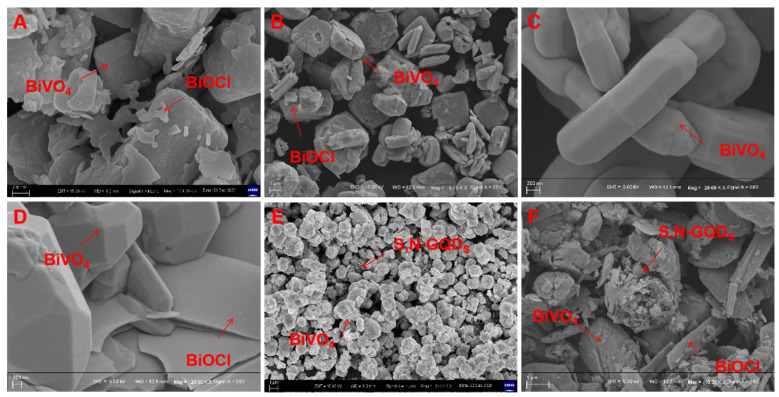
SEM images of BiVO_4_ samples processed for different treatment times (**A**–**C**), BiVO_4_/BiOCl (**D**), BiVO_4_/S,N-GQD_S_ (**E**) and BiVO_4_/BiOCl/S,N-GQD_S_ (**F**).

**Figure 4 micromachines-13-01604-f004:**
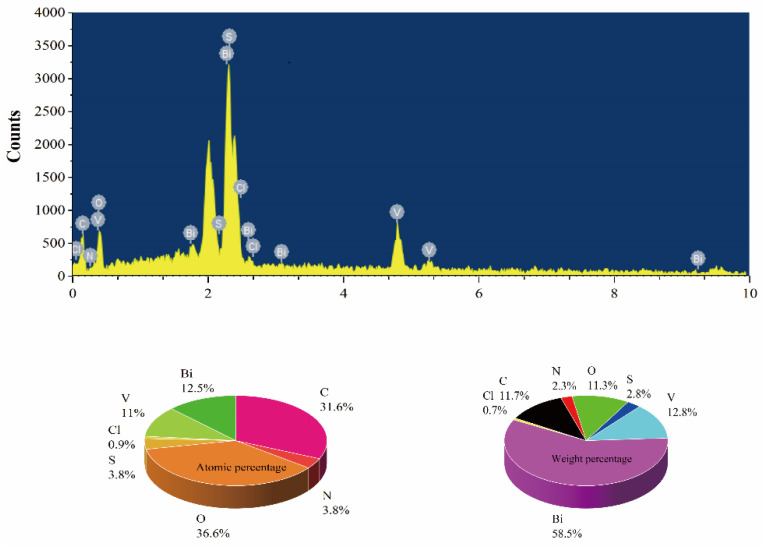
EDS energy spectrum of BiVO_4_/BiOCl/S,N-GQD_S_ (3%) sample.

**Figure 5 micromachines-13-01604-f005:**
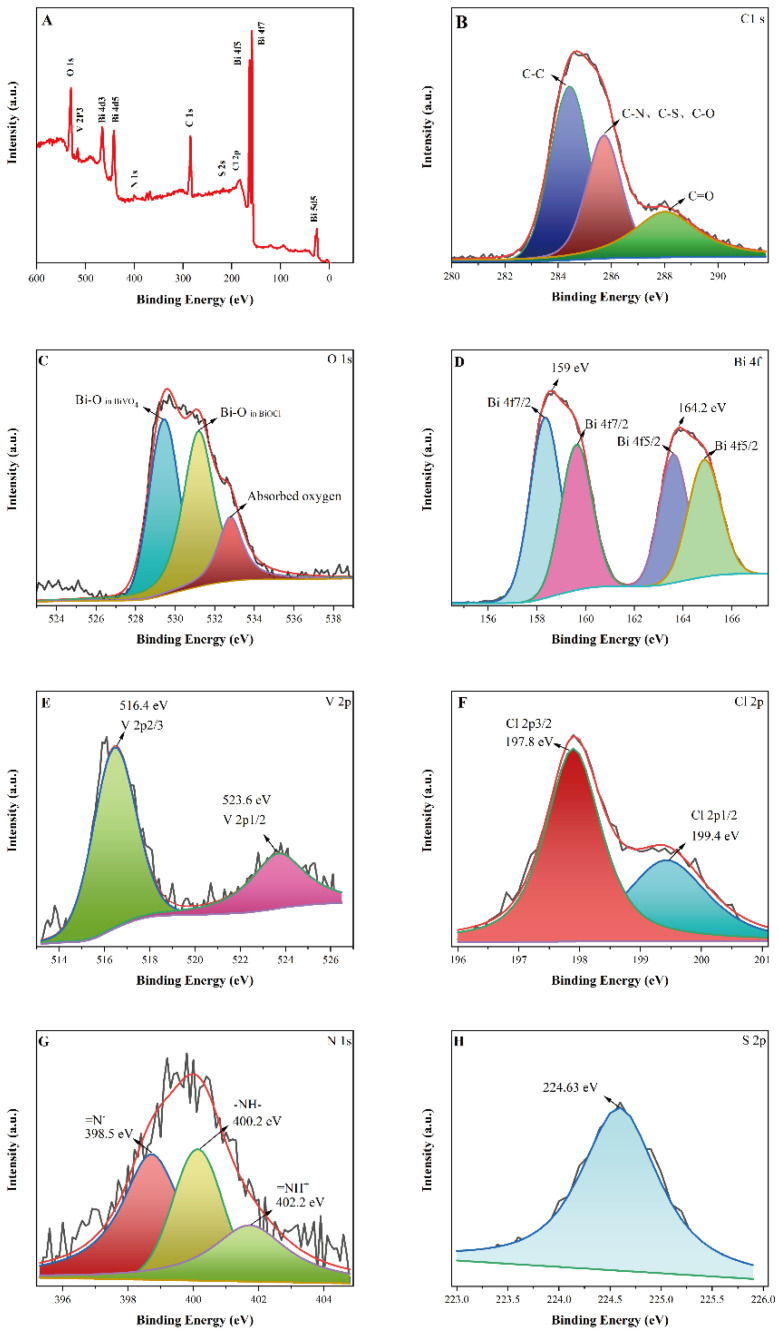
XPS patterns of BiVO_4_/BiOCl/S,N-GQD_S_ samples: full spectrum (**A**), C 1s (**B**), O 1s (**C**), Bi 4f (**D**), V 2p (**E**), Cl 2p (**F**), N 1s (**G**) and S 2p (**H**).

**Figure 6 micromachines-13-01604-f006:**
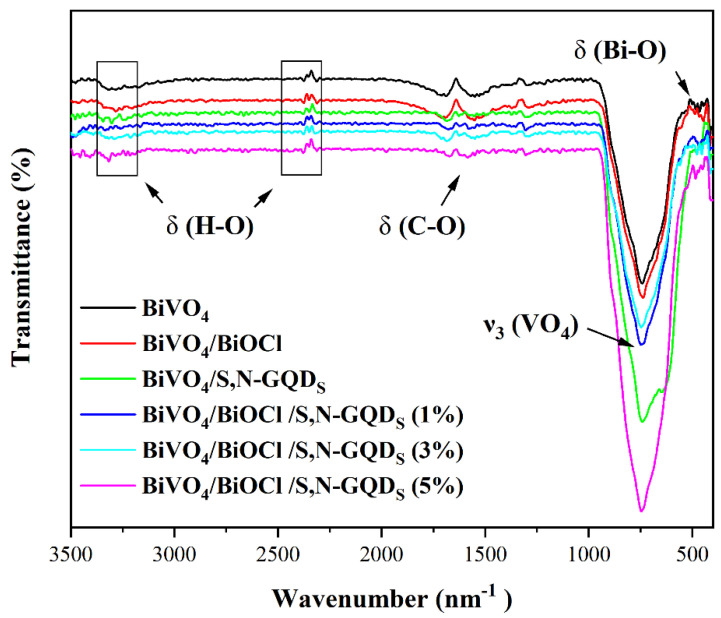
FT-IR spectra of BiVO_4_, BiVO_4_/BiOCl, BiVO_4_/S,N-GQD_S_ and BiVO_4_/BiOCl/S,N-GQD_S_ complex samples doped with different masses of S,N-GQD_S_.

**Figure 7 micromachines-13-01604-f007:**
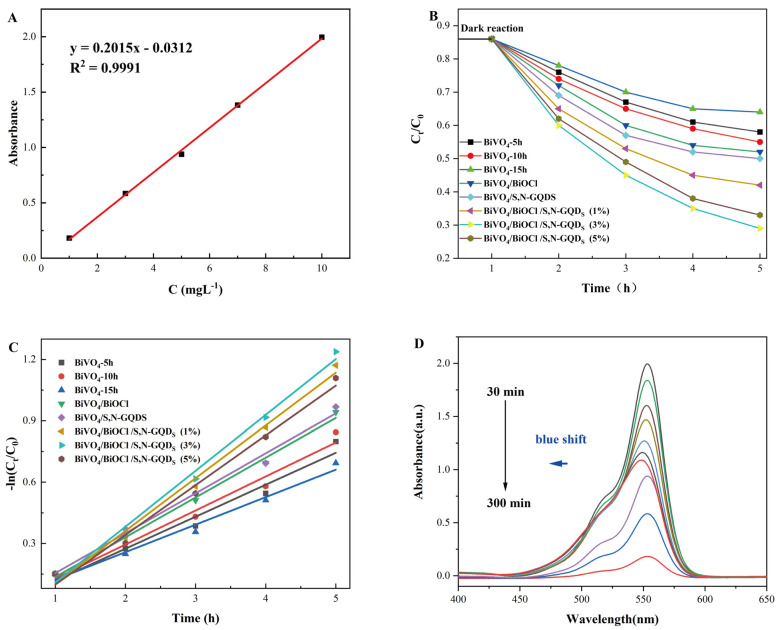
Experimental results of photocatalytic degradation: (**A**): standard curve of rhodamine B (RhB) solution; (**B**): photocatalytic degradation curve of prepared samples; (**C**): kinetic study of the sample in the degradation of RhB; (**D**): UV absorption spectrum of rhodamine B degraded by the BiVO_4_/BiOCl/S,N-GQD_S_ (3%) sample.

**Figure 8 micromachines-13-01604-f008:**
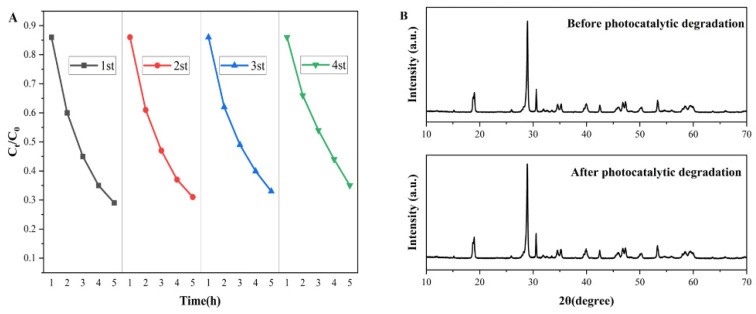
Photostability test plots of the cyclic photocatalytic degradation of RhB by the BiVO_4_/BiOCl/S,N-GQD_S_ sample (**A**) and XRD patterns before and after photocatalytic cycling experiments (**B**).

**Figure 9 micromachines-13-01604-f009:**
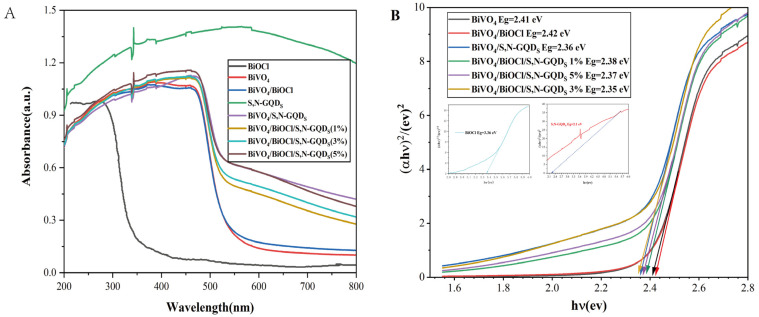
UV-vis diffuse reflectance spectra of different samples (**A**) and band-gap energy (**B**).

**Figure 10 micromachines-13-01604-f010:**
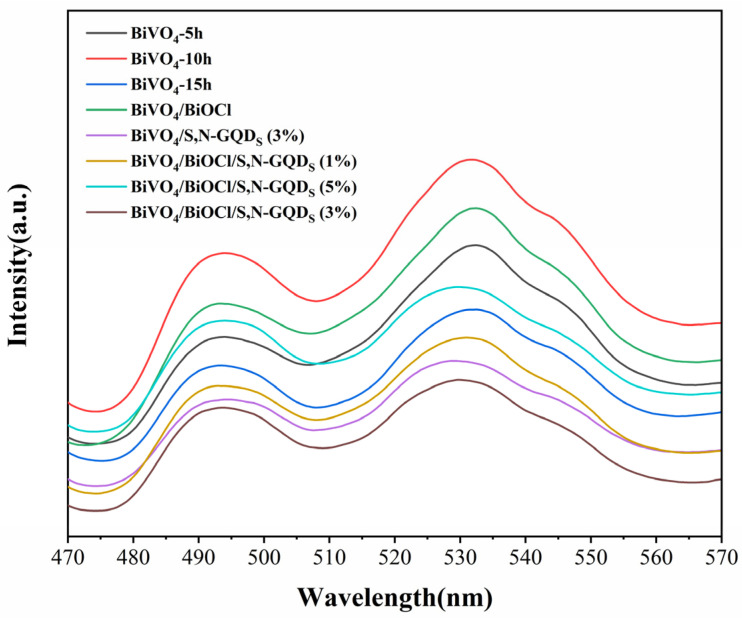
The PL spectrum of BiVO_4_ samples after different treatment times: BiVO_4_/BiOCl and BiVO_4_/S,N-GQD_S_ composites and BiVO_4_/BiOCl/S,N-GQD_S_ complex samples doped with different masses of S,N-GQD_S_.

**Figure 11 micromachines-13-01604-f011:**
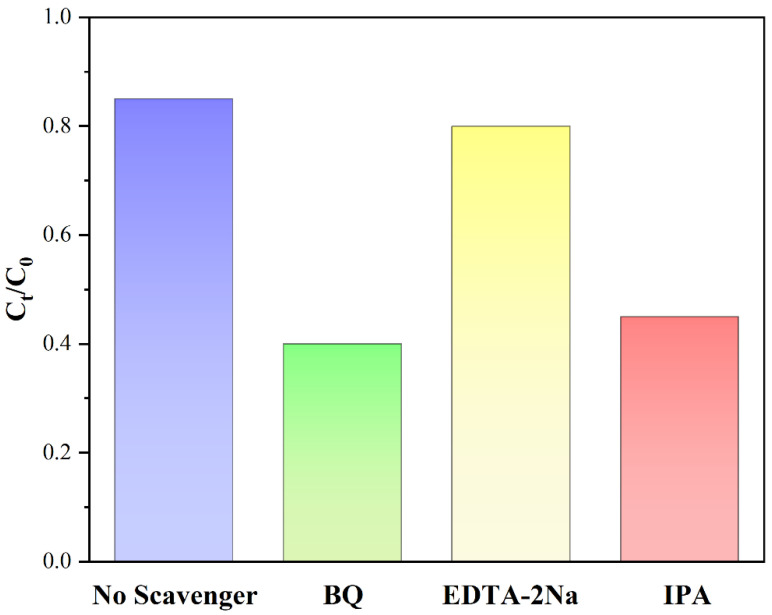
The results of the reactive-group-capture experiment for the photocatalytic degradation of RhB by the BiVO_4_/BiOCl/S,N-GQD_S_ (3%) sample.

**Figure 12 micromachines-13-01604-f012:**
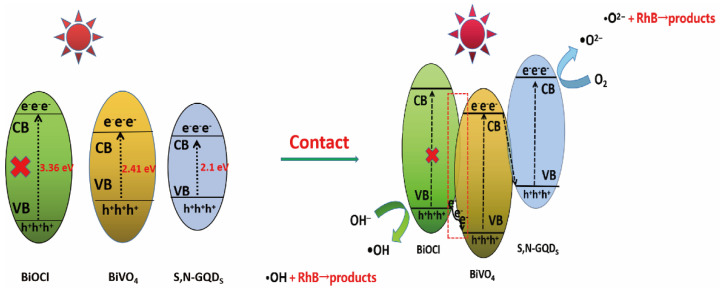
Schematic diagram of the mechanism of photocatalytic degradation of RhB by BiVO_4_/BiOCl/S,N-GQD_S_ samples.

**Table 1 micromachines-13-01604-t001:** Comparison of degradation performance of RhB by bivo_4_ based heterojunction.

Sample	m_photocatalyst_:m_RhB_	Illumination Source	Degradation Rate/Time	Reference
BiVO_4_	208:1	Halogen Lamp (500 W)	97.7%/6 h	[30]
BiVO_4_/BiOCl	125:1	Xenon Lamp (500 W)	93%/210 min	[31]
BiVO_4_/BiOCl/WO_3_	150:1	Xenon Lamp (350 W)	75%/3 h	[32]
BiVO_4_/TiO_2_	208:1	Xenon Lamp (350 W)	62%/6 h	[33]
BiVO_4_/BiOCl/S,N-GQD_S_	100:1	Xenon Lamp (350 W)	85%/5 h	-

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
