# Peer review of "Deep-Eutectic-Solvent-Assisted Synthesis of a Z-Scheme BiVO4/BiOCl/S,N-GQDS Heterojunction with Enhanced Photocatalytic Degradation Activity under Visible-Light Irradiation"

_micromachines, 2022, doi:10.3390/mi13101604_

Round 1

Reviewer 1 Report

The manuscript titled "Synthesis of a Z-Scheme BiVO4/BiOCl/S,N-GQDS Heterojunction with Enhanced Photocatalytic Degradation Activity under Visible Light Irradiation" is a well-structured paper that deals with the synthesis of a heterojunction that is employed for photocatalysis of rhodamine B.

Overall, the paper has a good format starting with material characterization and then exploring in detail the photocatalytic activities of the heterojunction providing optimization of the S,N-GQDS doping amount for better degradation efficiency and stability tests.

I recommend this manuscript to be published in Micromachines but firstly, it should undergo some minor corrections:

1. In the XRD section, the space group and the structure for both BiVO4 and BiOCl should be mentioned

2. Regarding the TEM (and consequently the SEM): no EDS maps are presented. As such it is very difficult to distinguish between the two materials in Fig. 2B. Similarly for SEM, it would be very good to show the EDS maps as they can present the homogenous distribution of the elements across the materials presented. Also no SAED is provided for the crystallinity of the materials

3. On another note, the scale bars and magnification are not clear enough for any of the SEM or TEM images

4. For sulphur XPS please present S 2p instead of S 2s

5. Figure 8B has Chinese letters and units on the y-axis, please correct and remove the units to be consistent with Fig. 1

6. Regarding the stability of the heterojunction after 5h: while the XRD is a good technique to show that the crystallinity remains unaffected, more data should be provided such as TEM for direct comparison

7. Finally, the catalytic properties of bismuth-containing materials have been explored a lot recently (e.g. https://doi.org/10.1021/acsanm.0c02860) Can the authors include such topic in the Introduction section?

Reviewer 2 Report

Title: Synthesis of a Z-Scheme BiVO4/BiOCl/S,N-GQDS Heterojunction with Enhanced Photocatalytic Degradation Activity under Visible Light Irradiation

1.      How will nitrogen present in GQDs preparation? Justify.

2.      S, N-GQDs is stable in HCl solution? How will the authors prepare the nanocomposites?

3.      In XRD analysis, there is no corresponding peak for BiOCl in BiVO4/BiOCl/S,N-GQDS composites. Why?

4.      The authors should give the clear TEM image for GQDs. In addition, the TEM scale bar should give the visible.

5.      Fig.7B, and Fig 8a is not correct. It should be in correct form of kinetics.

6.      Is there any bandgap of GQDs? Because it shows full UV and visible range in UV-DRS analysis. If correct, the authors should give some of the reference.  

7.      The manuscript contains grammatical errors and typographical mistakes. Eg. line 93, ‘kept in a Mafu furnace’

8.      Following double Z-scheme reference may include in the manuscript for more readable, Chemical Engineering Journal 423 (2021) 130076, Composites Part B: Engineering 234 (2022) 109726.

Reviewer 3 Report

I read carefully an interesting and comprehensive research article entitled Synthesis of a Z-Scheme BiVO4/BiOCl/S,N-GQDS Heterojunction with Enhanced Photocatalytic Degradation Activity under Visible Light Irradiation. The concept of the article is interesting and suitable for ‘Micromachines’. This manuscript is generally well written and clearly presented however still needs to address some comments, and thus require moderate revision to improve the quality of the manuscript.

1.      Title should modify which can describe the whole research work.

2.      Provide a nice graphical abstract representing the overview of the MS with key highlights.

3.      Abstract looks very general. Authors should mention the importance of research work briefly.

4.      In the introduction discuss briefly recent advances of photocatalytic degradation of organic contaminants in terms of ecological concerns.

5.      Why BiVO4/BiOCl/S,N-GQDS selected? As well as why selected RhB?

What about statistical analysis used in this study?

6.      The developed system has potential to degrade other contaminants?

7.      What about the degradation products of RhB are they toxic?

8.      Have authors utilized developed system for repeated use?

9.      Check the analytical properties of photocatalyst after repeated use

10.  Add one comparative table of the results obtained with the literature

11.  Techno Economic challenges and limitations of this system should be included? Add future research directions also.

12.  The conclusion of the study needs to be added with the specific output obtained from the study, it could be modified with precise outcomes with a take home message.

13.  Some English and grammar mistakes are present that need to be correct to improve the quality of the manuscript.

Round 2

Reviewer 2 Report

Accepted 

Author Response

Thank you very much for reviewing the manuscript

Reviewer 3 Report

I read carefully the revised version of the manuscript and still need some clarification before its acceptance

What about statistical analysis used in this study? If authors have not used any statistical analysis then the reliability of data is also questionable. How many times have you repeated  photocatalytic degradation experiments?

Add one comparative table of the results obtained with the literature which is recommended.

Author Response

(The authors gave the same response as above.)
